# From Program to Practice: Translating Energy Management in a Manufacturing Firm

**Mette Talseth Solnørdal ***  **and Elin Anita Nilsen**

School of Business and Economics, UiT The Arctic University of Norway, Pb 6050 Langnes,
9037 Tromsø, Norway; elin.nilsen@uit.no
**\*** Correspondence: mette.solnordal@uit.no; Tel.: +47-7764-6016

**Abstract:** A promising way to stimulate industrial energy efficiency is via energy management (EnM) practices. There is, however, limited knowledge on the implementation process of EnM in manufacturing firms. Aiming to fill this research gap, this study explores the implementation of a corporate environmental program in an incumbent firm and the ensuing emergence of EnM practices. Translation theory and the 'travel of management ideas' is used as a theoretical lens in this case study when analysing the process over a period of 10 years. Furthermore, based on a review and synthesis of prior studies, a 'best EnM practice' is developed and used as a baseline when assessing the EnM practices of the case firm. Building on this premise, we highlight four main findings: the pattern of translation dynamics, the key role of the energy manager during the implementation process, the abstraction level of the environmental program and, 'translation competence' as a new EnM practice. Managerial and policy implications, as well as avenues for further research, are provided based on these results.

**Keywords:** energy efficiency; energy management practices; translating management ideas; case study

## 1. Introduction

Increasing environmental degradation and risks from disasters have placed the mitigation of climate change and the emission of greenhouse gases (GHG) among the most pressing issues of the twenty-first century. The industrial sector accounts for a large proportion (37%) of the world's total energy consumption [1]. Therefore, increased industrial energy efficiency (EE) is an important means for sustainable development [2], and is essential to reach global sustainability targets such as the Paris Agreement [3] and the European 2030 climate and energy framework [4]. Manufacturing firms can increase their EE by implementing new technological measures in their production processes [5] that require less energy to perform the same functions [6], and by behavioural changes [7]. EE reduces energy costs [8] and increases productivity [9,10] and is positively related to firms' financial performance [11,12] and competitiveness [13]. Nonetheless, research has identified a wide range of barriers for industrial EE [14–16]. Hence, the manufacturing sector's full potential remains unexploited [17–19], leading to an 'EE gap' [20] which denotes the discrepancy between the theoretically optimal and current level of EE.

While the EE gap has mainly been addressed with relevance to technological innovations [21,22], it also consists of behavioural and managerial components, conceptualised as the 'extended EE gap' [7]. Thus, manufacturing firms' enhanced EE improvements and sustainable development [23] require that they accompany technological innovations with the implementation of energy management (EnM) [24–27]. EnM is thus a significant driver for increased EE in manufacturing firms [28,29] and requires academic attention.

Industrial EnM has been thematised in a number of publications, and comprehensively reviewed by e.g., Schulze et al. [30] and May et al. [26]. These reviews illustrate the broad variety in the

conceptual understanding of EnM in academic research. Lawrence et al. [25] describe EnM as the procedures in industrial firms addressing energy use to improve EE. Moreover, EnM is described as technical energy monitoring and measurement systems [31], and organizational systems for the continual improvement of energy performance [32]. Furthermore, EnM is considered a tool in helping firms overcome barriers to improving industrial EE [33,34]. Agencies such as the International Organisation for Standardisation (ISO) denote EnM as a standard, such as the ISO 50001 [35] energy management standards, whereas Sannö el al. [27] and Sa et al. [36] describe EnM as a program. EnM can also be incorporated in environmental policy programs at both national and regional levels [22,37]. Despite considering EnM from different perspectives, these studies present EnM as ideal recipes, at various abstraction levels, that firms should commit to for increasing their EE. Furthermore, in a comprehensive definition, EnM Schulze et al. [30] assert that 'EnM comprises the systematic activities, procedures and routines within an industrial company including the elements strategy/planning, implementation/operation, controlling, organisation and culture and involving both production and support processes, which aim to continuously reduce the company's energy consumption and its related energy costs'. This definition endorses the need for transforming EnM into EnM practices in the organisation, at the strategic, operational and human level.

It is worth mentioning, however, is that the contribution of EnM depends on firm-specific and contextual characteristics such as size, energy intensity and production type [7,27], and that these EnM recipes need to be given content and meaning according to the contextual setting of each firm. To this end, there is a broad variety in types of EnM practices [38] that firms can consider. Nonetheless, research shows that manufacturing firms often fail to implement EnM practices and that their EnM maturity level is generally low [8,25,32,39]. This suggests that firms face large challenges when implementing EnM.

In this emerging research field on EnM practices scholars have mainly focused on how to characterise effective and successful EnM practices [16,30,32,33,38,39]. Although these studies provide valuable information for describing and assessing the level of EnM practices in firms [27,36], they do not provide knowledge on the implementation process where EnM programs are transformed into specific EnM practices. This gap in the literature is addressed by Lawrence et al. [25] asserting that 'while barriers to and drivers for industrial EE have been investigated for many industries, there is a lack of studies of barriers to and drivers for EnM practices'. Hence, there is little knowledge on how firms adopt EnM [30] and align EnM practices with firms' core business and strategic agendas [40]. Models to support industrial managers in successfully implementing EnM are also lacking [38]. Furthermore, with the exception of Sannö et al. [27], few empirical studies analyse the implementation of in-house EnM programs in multinational companies (MNC). Hence, there are several calls for more knowledge on the implementation of EnM programs in MNC.

A research tradition that pays attention to the transformation from programs to practices is the 'translation perspective' [41], located within the framework of Scandinavian institutionalism [42–44]. Here, the inherent premise is that implementation is closely associated with the translation of management ideas and models [41,44]. A main focus in this approach has been on variations in how new versions of organisational ideas are translated in the local context. Empirical research has identified that translation takes place in accordance with translation or editing rules, and that 'good translations' foster successful results [45–47]. As such, the outcome of a successful implementation process can be observed in the materialisation of, for instance, new routines and practices. According to this approach one would assume that EnM goes through a transformation from a vague idea to concrete practices as part of implementation. Furthermore, scholars assert that how this transformation is approached with regard to translation will affect if and how the EnM program materialises as definite EnM practices in the organisation.

Using this framework, this study aims to meet these calls by considering the implementation of EnM as a process of translation. We ask: *What is the relationship between the translation process of a corporate environmental program and the successful materialisation of EnM practices in a manufacturing firm*?

The case study is new in the sense that it explores retrospectively over a period of 10 years (2004–2014) the implementation of a corporate environmental program, here called 'EcoFuture'. The data are qualitatively collected in 'Pharma', a subsidiary of an MNC. Through EcoFuture, the global management of the MNC pledged to integrate long-term sustainability objectives into its global business strategy and the business strategies of all subsidiaries.

As such, the study contributes to the EnM literature with new knowledge on the relevance of contextual factors and the role of key translators in reinterpreting and giving the environmental program content in the firm setting. In addition, the results add to our limited knowledge of corporate EnM and EnM practices at the firm level. Moreover, the results indicate the potential of the translation framework in research on EnM, and suggests its use as a 'tool' to gain translation competence to facilitate successful implementation of EnM. This will benefit managers by highlighting specific competences and actions for successfully implementing EnM and obtain enhanced EE. Regarding policy implications, the results point to the relevance of the design of environmental policy programs for effectively stimulating industrial EE.

The remainder of the paper is structured as follows. Section 2 describes the theoretical framework of the study, including a literature review of EnM practices and a presentation of the translation theory. The research methods and data collection are outlined in Section 3. Section 4 presents the empirical analysis and the results of the study. In Section 5 we offer a discussion of the research findings. Finally, in Section 6, we provide a conclusion where the limitations, implications and avenues for future research are highlighted.

## 2. Theoretical Framework

### 2.1. EnM Practices

The objective of EnM is to continuously increase manufacturing firms' EE in production processes [30]. From a wider perspective, energy management contributes to the environmental transition and sustainable development of manufacturing firms by incorporating economic and environmental factors into overall business strategies [40]. In the literature, EnM is defined as 'the strategy of meeting energy demand when and where it is needed' [48] and 'procedures for strategic work on energy [21]. Moreover, Bunse et al. [49] define 'EnM in production as including control, monitoring, and improvement activities for EE', and Ates and Durakbasa [39] emphasise that 'EnM is considered a combination of EE activities, techniques and management of related processes which result in lower energy cost and $CO_2$ emissions'. The comprehensive definition by Schulze et al. [30] (see intro) endorses that EnM needs to be operationalised as EnM practices.

EnM practices include a broad range both technical and managerial elements [38,50], by which the spillover effect from general management practices to EE has been investigated [22]. Scholars in the field also have put forward several frameworks for characterising and assessing EnM practices in manufacturing firms, such as: Minimum requirements, Success factors for in-house EnM, EnM Matrix and Maturity models. The frameworks are described in the following text, while the EnM practices considered in these frameworks and in other relevant studies are depicted in Table 1. We identified the reviewed articles by searching for 'energy management practices' in Google Scholar and through manual screening of cross-references. The list of articles is not exhaustive but provides an overview of the EnM practices considered relevant for EE improvements in recent research.

The Minimum requirements model is a basic framework for assessing EnM in manufacturing firms [32,39], by limiting the analysis to evaluate to what extent EnM is put into practice. Based on this model, a survey study covering 304 Danish industrial firms, concluded that only between 3% and 14% of the firms practiced EnM [32]. A multiple case study in Turkey found that only 22% of the surveyed companies practiced EnM [39]. Furthermore, based on a review of empirical research on industrial EnM in Sweden, Johansson and Thollander [33] propose a framework of ten success factors for efficient in-house EnM. This framework is used in empirical studies to monitor the adoption of an

EnM program in an MNC and assesses the performance of these key elements [27]. The five level EnM matrix framework is a more elaborated framework that consists of six organisational issues related to EnM [51]. Gordic et al. [50] used this matrix when analysing the procedure for developing EnM in a Serbian car producing company. The EnM matrix is often combined with the Maturity model which is a relevant framework for evaluating firms' ability to manage energy [36,52].

The Maturity model contributes to a better understanding of suitable EnM configurations and the required steps to establish EnM practices according to firms' energy strategies [36]. Empirical studies from Sweden applying the Maturity model shows that the maturity level of firms is relatively low [36,53]. The EnM maturity and matrix models have several similarities in addressing firms' sophistication level of EnM practices and take into account more detailed descriptions of activities considered as EnM practices. The most comprehensive presentation of EnM practices is proposed by Trianni et al. [38]. Based on a literature review they develop a reference list of 58 EnM practices that are specified and can be used as a baseline for benchmarking firms' implementation of EnM practices.

Thus, the literature emphasises the importance of adopting EnM practices within an organisation and proposes assessment models with definite practices that characterise effective EnM. The framework of EnM practices depicted in Table 1 builds on this literature and serves as a point of reference when assessing the EnM practices of Pharma. In the following this is referred to as 'best EnM practice'. Table 1 synthesises the findings in the literature on how to characterise successful EnM practices. These theoretical 'best EnM practices' assert the need to adopt both management routines and organisational structures, in addition to competence-enhancing activities. Considering environmental leadership practices, it is essential that top managers support and are committed to the environmental agenda and formulate long-term environmental strategies and goals. Management practices should also focus on employee involvement and motivation. Furthermore, successful EnM practices depend on dedicated personnel working on energy matters and a clear allocation of responsibility. Additionally, performance measurement systems are essential in controlling, monitoring, and planning energy consumption against strategic targets, thus allowing for effective information assimilation and reporting to management and operational personnel. Competence is positively related to environmental awareness; hence, education and training are outlined as important ways to improve internal energy performance. Studies also assert that EnM should be reflected in a firm's investment decision processes and plans by prioritising environmental business objectives and allocating resources to EE projects. Firm characteristics and operations, such as production processes, innovation, and R&D focus are also found to influence EnM. Worth to mention, however, is that although some studies include energy costs and external factors as EnM practices, they are not included as such in this study since they are not considered to be part of organisational structures or routines.

**Table 1.** Theoretical 'best energy management (EnM) practices' based on a synthesis of scholarly articles.

| Categories | Energy Management Practices (EnM Practices) | Thollander and Ottosson [8] | Martin et al. [11] | Brunke et al. [16] | Sannö et al. [27] | Schulze et al. [30] | Christoffersen et al. [32] | Johansson and Thollander [33] | Ates and Durakbasa [39] | Gordić et al. [50] | Jovanović and Filipović [52] | Sa et al. [53] |
|---|---|---|---|---|---|---|---|---|---|---|---|---|
| Management and Environmental leadership | Top management support and awareness of energy issues | | X | X | X | | X | X | X | | X | |
| | Energy strategy (policy), planning, and targets | X | X | X | X | X | | X | X | X | X | X |
| | Employee involvement, communication, motivations and incentives | | | | X | X | | X | | X | X | X |
| Energy manager and Organisational structures | Energy manager and the strategic positioning of the energy manager in the organisation | | | X | X | X | X | X | X | X | X | X |
| Performance measurements | Information systems, energy audits, sub-metering, controlling and monitoring | X | X | X | X | X | | X | | X | X | X |
| Competence | Staff awareness, education, and training (culture) | | | | X | X | | X | | X | | X |
| Investment decision | Investment and pay-off criteria, and allocation of energy costs | X | X | X | | X | | | | X | | X |
| Firm characteristics | Energy prices and competitiveness | | X | | | | X | | X | | | |
| | Firm characteristics | | | | | | X | | X | | | |
| | Operations and production processes | | | | | | | | | | | |
| | Innovation and R & D focus | | X | | | | | | | | | |
| External factors | Policies and regulations | | X | | | | X | | X | | | |
| | External relations | | | | | | X | | X | | | |

*2.2. Translation Theory*

In exploring the materialisation of EnM practices in Pharma, this study considers EcoFuture a management idea and analyses the firm's internal translation process of the idea. Sahlin-Andersson [43,44] defines management ideas as successful models that provide solutions to pressing problems in different contexts and at different points in time. They can further be described as social and legitimised norms for how an efficient organisation should appear regarding structural arrangements, procedures, and routines [54], such as codes of ethics [55], lean management [56] and reputation management [57]. When travelling between settings, ideas are conceived as immaterial accounts that are dis-embedded from their original contexts in terms of time, space, and location [43,57]. Hence, when an idea is re-embedded in a new setting, it must be translated and recontextualised [55]. As the translation process may include a broad range of translators, including government personnel, managing directors, middle managers, researchers, consultants, and operational personnel [58,59], there are numerous ways of translating the idea. However, empirical research has identified that translation takes place in accordance with translation or editing rules [55,56,60], that are applied more or less deliberately as a chosen strategy [47]. The outcome can be witnessed through changes in e.g., the mindset of individuals, formal documents, and the enactment of new practices [57].

Scholars have conceptualised the translation process through, for example, translation modes and rules [47,57], abstraction levels [61,62], and translation processes [63]. The theoretical framework in this study builds mainly on editing rules [43,44], which are apt for analysing the translation of broad ideas into local workplace practices. The first editing rule concerns the context and the process by which the idea is made appropriate for the local setting. In recontextualising the idea, organisational members add time, space, and sector-bounded features and make it relevant to the local setting. In this study, emphasis is placed on regulative and normative sector-bounded features and macroeconomic issues. The second editing rule concerns the formulation and labelling of an idea. The focus is on how the idea is formed so that it is deemed appropriate in the new context by discarding and adding elements to the idea [57]. Relabelling offers explanations for why an idea is successful and allows an idea to 'seem different but familiar' [56]. This rule is therefore relevant when analysing how EcoFuture was formulated when communicated in the organisation. The third rule relates to the plot of the story or the rules of logic. Sahlin-Andersson [43] describes this rule as the rationale behind the idea, in which 'explanations are given as to why a certain development has taken place'. Doorewaard and Van Bijsterveld [63] describe this as a power-based process in which the actors 'continuously reshape the elements of this process by confronting their own ideas with those of others and with existing organisational practices'. Prior empirical studies provide different examples of how to operationalise this editing rule [56,60]. Here, the analysis includes the translators' endeavours in fitting EcoFuture in the organisation setting through linking it to a known and acceptable internal logic, and thereby stimulating engagement among operational personnel. The theoretical framework is summarised in Figure 1.

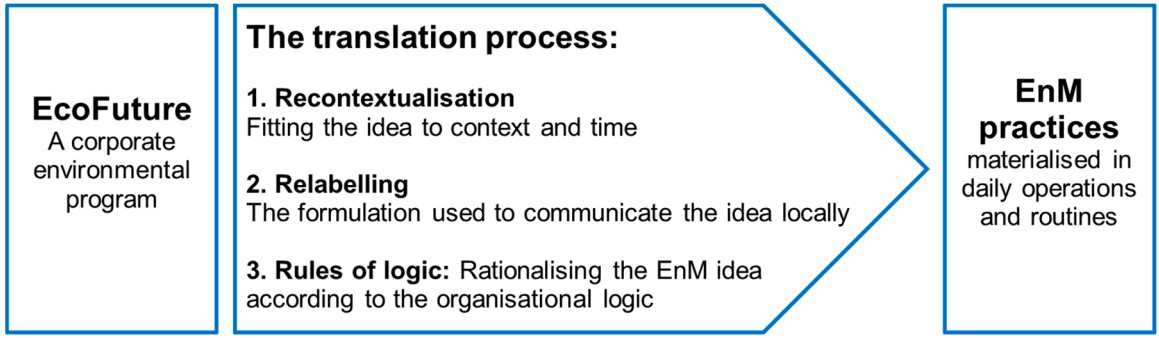

**Figure 1.** From program to practice: the translation of a corporate environmental program to EnM practices in a manufacturing firm.

The figure depicts how the original idea travels from the corporate to the firm level and illustrates the translation of the environmental program. It is important to point out that time is essential in the translation process, which is not captured by this model.

## 3. Methods

### 3.1. Case study Research Design

We selected a retrospective longitudinal case study as a research design when exploring the implementation of a corporate environmental program in a manufacturing firm. The case analysed here is the implementation process of a corporate environmental program in a context-bounded setting. Ragin [64] supports this understanding of a case and 'consider cases not as empirical units or theoretical categories, but as products of basic research operations'. Through the research design we aim to present a detailed understanding of the process, which can be found in qualitative data sources [65]. Furthermore, we chose the single case study approach because of its ability to provide rich and detailed data [66] on the process and the contextual setting in which the implementation occurred. Such contextual understanding is critical for the translation process and the organisational logics behind the emergence of EnM practices. Case study designs are accordingly often used in translation research [55,56,58] and in empirical studies on the implementation of environmental programs [67]. A common criticism to case study design is its inability to make statistically valid generalisations beyond the particular case. In the case of study research, however, the aim is rather to contribute to analytical generalisation [66], by analysing the contextual description of the behaviours and actions that are embedded in the empirical context through theoretical lenses [68]. Through the process of confirming, developing or extending a theory's area of use, analytical generalisation is achieved.

### 3.2. The Case and the Empirical Context

The case analysed in this study is the implementation process of a corporate environmental program of an MNC, here named EcoFuture. The MNC operates in a multitude of sectors, employing more than 300,000 people in over 180 countries. EcoFuture has run from 2004 and aims to integrate long-term sustainable objectives into the core of the MNC's business strategy by: (1) increasing its investment in clean R&D (cleaner technologies); (2) increasing revenue from EcoFuture products, defined as products and services that provide significant and measurable environmental performance advantages to customers; (3) reducing greenhouse gas (GHG) emissions and their intensity, along with improving EE; and (4) informing the public. At the corporate level, EcoFuture is mainly based on quantifiable environmental targets and without any detailed instructions on how it should be operationalised at the local level in the subsidiaries. The abstraction level of the program was accordingly rather high [62]. The program was globally revised in 2007, 2009, 2010, and 2014.

The empirical context selected for this study is a pharmaceutical firm in Norway with about 100 employees—Pharma. The firm specialises in producing drug substances for contrast agents used in medical imaging. Most of the firm's economic activity relates to two products, for which it holds a significant market share. The firm thus produces mainly commodity-type products of high volume for global markets, and high capital investments in facilities, which are typical in the broad-process industrial sector [40]. Pharma was purchased by the MNC in 2004 and became subject to EcoFuture. Hence, the decision to implement the program was at corporate level and not within the hands of the firm. Nonetheless, Pharma had to find means to fit the program to the local context, which is the process explored in this study.

### 3.3. Data Collection

The case is selected purposefully [69], due to Pharma's impressive EE improvements over time and recommendations from various sources such as environmental NGOs, energy mangers in other companies and public agencies. The case was therefore considered information-rich and adequate

to answer the research question [70]. Primary data were collected through interviews, observation, firm internal manuals, and other documents during three company visits over a period of four months in 2014. Secondary data were also collected such as annual reports, conference presentations and press articles. The data collection started with a meeting with the energy manager at Pharma. At this first meeting, the overall objectives of the research project were discussed. The second company visit included a guided tour of the site and observations in some of the factories, a firm presentation, and an interview with the firm's top management. Additionally, we were given access to internal documents such as investment prospects, project descriptions, and project manuals. We also collected secondary information from press articles, marketing brochures, conference presentations, and annual corporate reports. This information was used to prepare for new interviews. During the third company visit, semi-structured interviews were carried out with key informants. The main purpose of the interviews was to understand informants' perceptions of the firm's operations, including production processes, R&D, technology implementation, and decision processes and practices for employee involvement and training. Additionally, they were asked about their understanding of EcoFuture and the integration of EnM into the firm's daily operations and activity. Following Eisenhardt and Graebner [71], data were collected from highly knowledgeable informants who viewed the focal phenomena from diverse perspectives. The energy manager helped identify key informants representing a cross-section of the organisation that were engaged in and/or affected by EcoFuture, including the director of health, safety, and environment (HSE), the energy manager, project managers, site managers, operating personnel, R&D staff, and members of the EcoFuture team. This cross-sectional sample allowed for a comprehensive understanding of the translation process and the EnM practices within Pharma. For practical reasons, some of the interviews were performed in groups of 2–3 interviewees. In total, nine key informants were interviewed. Two of the informants, considered particularly knowledgeable, were interviewed more than once. Hence, during the period, eight interviews were conducted, each lasting 1–2.5 h. All interviews were fully transcribed.

### 3.4. Method of Analysis

The analysis of the empirical material rests on the framework of directed content analysis [72]. The goal of a directed approach to content analysis is to validate or extend conceptually a theoretical framework or theory, and one might categorise this as a deductive approach. In our case we used translation theory, and more specifically the editing rules, as key concepts to create coding categories to guide the reconstruction of the material. As such, we also used translation theory to help focus the research question. Such theoretical conceptualisation of the empirical data is particularly suitable for case studies [70].

The analysis started with a chronological presentation of the material. This was done by triangulating data from the interviews, annual reports, conference presentations and press articles. The chronological presentation was then followed by an effort to code the material according to the editing rules described in the theoretical framework (Figure 1). Moreover, the researchers analysed the EnM practices in Pharma by using data from the interviews and observations at the site, and from internal documents and manuals describing routines and investment projects. NVivo, a qualitative analysis software product, was used to store the material and as a support for the coding. The operationalisation of the editing rules allowed the identification of shifts in intensity of the translation process that indicated patterns in the implementation of EcoFuture. Subsequently, we categorised the material into three periods: 2005–2007, 2008–2010, and 2011–2014. The first phase was mainly recognised by the contextualisation of EcoFuture to the local context of the firm. In the second phase, external economic shocks called for larger organisational changes and stimulated managerial efforts to implement EcoFuture within the organisation. The third phase was recognised by the efforts of the energy manger in rationalising EcoFuture into EnM practices. We named these phases Complacency, Urgency and Maturity, drawing the attention to some main characteristics regarding translation during these periods. As such, temporal bracketing was applied as an analytical

strategy [65] by which the data are decomposed into successive periods. This strategy permitted the creation of comparative units of analysis [73] and helped us analyse how the translation of EcoFuture unfolded over time. Furthermore, this strategy allowed the inclusion of time as an additional dimension to the theoretical framework (Figure 1). Finally, the researchers assessed the outcome of the translation process, in terms of successful materialisation of EnM practices, by benchmarking Pharma's EnM practices with the theoretical best EnM practices described in Table 1.

There are some limitations related to the research design, for instance regarding the context bounded character of the data provided. This also implies challenges in making causal connections between actions and results in single case studies. Nevertheless, several strategies have been applied to ensure the rigor and reliability of the study. First, we used a research framework, derived from translation theory, to guide data collection and the analysis. Furthermore, the analysis is based on multiple data sources, and the triangulation of the material allowed us to develop an understanding of the translation process over time and reduce the intrinsic biases in empirical data.

## 4. Results

The presentation of results is structured according to the three chronological time periods described in Section 3.4—Complacency, Urgency and Maturity. By following the translation across these periods, the analysis illustrates how EcoFuture goes through a maturity process before unfolding as EnM practices. The results related to the translation process are summarised in Table 2 below, while the outcome of the translation process in terms of EnM practices is discussed in Section 4.4.

**Table 2.** Translation process of the idea according to translation rules and period.

| Period | External Factors | Translation Rules | | | Key Translators |
|---|---|---|---|---|---|
| | **Context** | **Recontextualisation** | **Relabelling** | **Rules of Logic** | |
| | Contextual factors external to the firm affecting the translation of the idea | Fitting the idea to the context and time | Formulations used in communicating the idea locally | Rationalising the idea according to the inherent organisational logic | |
| Complacency | EcoFuture launched (2005) as a corporate environmental program EcoFuture targets include: Increase investment in R&D of cleaner technologies Increase revenues from products and services that provide environmental performance and advantages to customers Reduce GHG emissions and improve the EE of the firm's operations | EcoFuture was contextualised according to national environmental regulation, sector bounded technological and regulative features, in addition to limited access to capital, and translated as an EE program | - | The editors were focusing on synergies between environmental and efficiency objectives—'picking the low-hanging fruit' | Top managers |
| Urgency | EcoFuture is revised (2009 and 2010) with new and increased targets Other external chocks: Global financial crisis Increased global competition Establishment of an industrial network for sustainable process industry | The program was locally contextualised as an EnM and organisation development program, with focus on productivity | The program was relabelled 'Smart Growth', including emphasis on energy efficiency | Economic rationale: all EnM investments should be economically feasible | Top managers |
| Maturity | EcoFuture is revised (2014) with new and increased targets | - | - | Economic rationale: all energy investments should be economically feasible Rationalising the program by integrating the technological complexity of the production processes, organisational resistance to change, and existing organisational integration of energy issues | Energy manager |

*4.1. Period 1: Complacency*

The first period is mainly recognised by the top managers' initial efforts to fit EcoFuture into the contextual setting of the firm. Normally, the translation process begins with the decision to adopt the idea [74]. In this case, however, the adoption decision was made at the corporate level, and the local managers had to find ways to fit EcoFuture to the local firm setting. Hence, the top managers contextualised EcoFuture according to regulative and technological sector-bounded features. The analysis points to the impact of pharmaceutical acts regulating lengthy procedures and product certificates before commercialising new drugs. In addition, licences and emission permits by the national environmental authorities affecting the firm's production figures also played a role in this contextualisation process. Furthermore, Pharma was characterised as having a few commodity-type products, continuous production in high volumes, extensive capital binding in existing production equipment and facilities, and limited access to investment capital. The following interviewee quote illustrates how EcoFuture imposed new environmental demands on the firm without the backing of additional resources:

> [Y]ou don't get any money for doing this. The environmental investments compete on equal terms with any other investment project. They say that we have to reduce the energy consumption, but they don't say 'Here, you have money to do it'. It doesn't work that way. The environmental projects have to enter ordinary budgets. That is tough!

These regulative and technological features prompted the managers to search for a way to implement EcoFuture without triggering significant operational changes or large investments. Subsequently, EcoFuture was translated to fit the extant portfolio of products and production processes by contextualising it as an EE program. During this first period, Pharma had significant potential for EE improvements, which were realised via minor adjustments and 'picking the low-hanging fruits'. Moreover, only the top managers were preoccupied with EcoFuture and no significant efforts were made to implement the program within the entire organisation. Such situations in which an idea resides high in the hierarchy being decoupled from organisational practice, is described as 'isolation' by Røvik [74], and casued the further travel of EcoFuture to stagnate. The analysis suggests a complacent translation of EcoFuture during this period, in the sense that translators in Pharma did not see the relevance or need to take large environmental steps and putting efforts into translating EcoFuture.

*4.2. Period 2: Urgency*

This second period is recognised by external economic shocks and changes in the institutional environment that boosted the relevance of EcoFuture. With this as a backdrop, the translation of the program intensified, as the top managers started to recontextualise and relabel EcoFuture with the aim to implement the program within the organisation. Indeed, the global financial crisis of 2008 led to reduced sales, price drops, and production overcapacity. Moreover, the patent protection on Pharma's products expired during this period, leading to increased competition from generic drug manufacturers. EcoFuture was also revised at the corporate level and the environmental targets amplified. As the firm had already taken advantage of the 'low-hanging fruits', greater organisational involvement was now needed, and it became necessary to rethink the strategy and organising of the program.

Economic considerations were prominent when top managers translated the program during this period. Furthermore, in translating EcoFuture into an internally understandable concept, the program was relabelled as a productivity program named 'Smart Growth', with an emphasis on EnM. The use of familiar rhetoric is an efficient means to avoid organisational resistance towards an idea [55]. Productivity was a well-established concept within the organisation. EcoFuture and EnM were, hence, fitted to the local setting by having productivity as a primary objective and taking advantage of the synergies between increased productivity and EE. This approach is illustrated by an interviewee stating that the '*environmental benefits are a spin-off of productivity*'.

The intensification of the translation process also involved the restructuring of the organisation and technical improvements. Furthermore, new EnM practices started to materialise such as energy audits, the systematic monitoring of main energy streams, and the measurement of total energy consumption, all of which are essential for attaining correct information, reporting, transparency, and constructing a shared course of action in strategic energy planning [75]. In addition, the operational personnel started to engage in EE. As illustrated by the following interviewee quote, this upscaling of the program can be considered an important premise for succeeding with energy improvements:

> [F]rom then, the program changed from being an energy-saving program that only some were engaged in to become a factory productivity program. This is one of the success criteria.

Contextual changes characterised this period. The analysis indicates that these changes created an urge to act in the organisation [74] that reactivated the relevance of EcoFuture. Hence, in contrast to the complacency described in the first period, the idea was now translated as an EnM and productivity program. As a result, the initially abstract idea now started to materialise further in the organisation. This could be observed in both the relabelling of EcoFuture and the emergence of EnM practices.

### 4.3. Period 3: Maturity

During the first two periods, EcoFuture was translated into an EnM program after being recontextualised and relabelled primarily by the top managers. In contrast, this third period is recognised by the further materialisation of new EnM practices. The analysis now reveals the energy manager as an important translator. This includes his efforts in aligning EnM with internal organisational agendas and structures, such as the technological complexity of the production processes, organisational resistance to change, organisational integration of energy issues, and the rationale of economic feasibility. By applying 'rules of logic' as a translation tool, the energy manager stimulated the development of accepted and legitimate EnM practices.

First, we find that the rationale of economic feasibility was used as a way of framing the implementation of the EnM program. In general, investments arise due to new technology, machinery wear-out, and changes in market demand, prices, and legal requirements. Investments are, thus, necessary to remain competitive in changing environments. The analysis indicates that the principle of 'good business' was an overarching logic in Pharma and thereby strongly affecting the internal investment decision processes and reporting schemes. Good business is understood as investments that are economically feasible in the short term and ideally lead to long-term sustainability. Indeed, all investments were based on economic considerations within which environmental improvements were deemed positive spin-offs. This approach is exemplified in the following interviewee quote:

> We don't really think that our first priority is to save the environment. However, we include it in productivity. We will show you how we plan to become CO2 neutral . . . It is, however, not a target in itself for this factory, even though there are some demands for us to become CO2 neutral.

The significance of the economic logic is also apparent in the way investment projects were evaluated and prioritised. The typical required payback time for investments was 2–3 years. Prior studies have noted that it is a challenge for energy projects to comply with such short payback demands, thus causing a significant barrier for industrial EE improvements [76,77]. Furthermore, all investment projects were categorised and rated according to compliance, health, safety and environment (HSE), maintenance, and productivity. The following interviewee quote illustrates the limited priority given to environmental objectives:

> Legislative compliance, HSE, and maintenance have priority before the green projects (because the green projects are sometimes productivity-related), and we don't get green projects through just because they are green.

The firm had organisational structures and routines for assimilating information and reporting to management and the organisation about energy consumption. Nonetheless, to gain support for energy-related projects during investment decision processes, energy improvements had to be argued and rationalised according to the extant economic logic and culture of the firm. Hence, arguments based on compliance, HSE, maintenance, and productivity had to be integrated and highlighted to win through with environmental projects. These strategies of searching for and aligning economic and environmental objectives are important for rationalising the EnM program according to the organisational logic. Nonetheless, these strategies contradict with recommendations from prior studies asserting that earmarked funding for environmental investments is a significant driver for EE [76]. Moreover, Sandberg and Söderström [78] state that the priorities made during investment decision processes depend on the culture of the firm. In other words, the strong economic rationale can be considered a cultural factor affecting the rationalisation of the EnM program in Pharma.

In this situation, the energy manager had seniority and operational experience, which provided him with in-depth knowledge about the factory and production processes. The analysis suggests that he took advantage of this experience and previously established energy monitoring practices -which provided him with accurate information on energy consumption, production bottlenecks, and investment opportunities—when rationalising the EnM program according to the logic of 'economic feasibility'. Radaelli and Sitton-Kent [59] describe several channels that middle managers can use in this translation process. Here, the energy manager worked proactively and used formal and informal arenas to continuously sell the EnM idea, manage conversations, and align diverging interests among organisational units and members.

Pharma had developed a complex production infrastructure over decades. Any changes in the production processes, including energy improvements, required not only excellent engineering skills, but also comprehensive and in-depth knowledge about the factory. External stakeholders were considered by Pharma to lack such detailed knowledge; thus, the innovation processes in the firm depended on the skills and creativity of the employees. Consequently, Pharma's EnM practices were largely determined by the employees' motivation and competencies in EnM. This finding coincides with prior studies underlining the role of operational personnel in process innovation and EnM [25] and the significance of employee motivation regarding EnM [31]. Hence, in addition to technological complexity, the translation of a management ideas depends on translators' ability to mitigate organisational resistance to change [60]. Such resistance was also experienced in Pharma, with the following interviewee quote illustrating the challenge of motivating employees to work collectively with EnM:

*[W]e have some examples [in which] we have completed projects in one area where the savings have been in another area, and then we have met a lot of resistance in the area in which the change has taken place.*

The integration of energy issues in organisational structures, or lack of such integration, was also part of the translation process. Pharma had systems and routines for controlling and monitoring the largest energy flows, however the analysis indicates that energy issues were only formally integrated in selected areas of the organisation. This restricted integration can be exemplified by the firm's use of key performance indicators (KPIs). KPI is a management tool developed to motivate employees in a preferred direction and is a recommended EnM practice [33]. Nonetheless, the use of KPI can also hamper change processes and compromise EnM practices [24], by stimulating other operational activities and investments. In Pharma, KPI was customised to the main activity of each unit. Hence, only the units with extensive energy consumption had energy use integrated into their KPI. Consequently, this use of KPI provided most employees with limited incentives for engaging with the firm's total energy consumption.

However, to motivate operational personnel to work collectively with EnM, Pharma provided several competence-enhancing schemes and activities such as internal training programs and education support, which are found to be significant drivers for industrial EE [28,79]. The following interviewee

quote illustrates how the benefits of energy improvements at the individual level were identified and communicated to provide motivation to make energy improvements:

> *The best projects are carried out when the area in which the change takes place [obtains] good benefits from the change. So, being very strategically smart, when you create projects, you have to find something that the department [in which] the change [takes place] benefits from.*

The energy manger also formed alliances with external stakeholders. The use of an external network is a well-known strategy for enhancing EE by exchanging knowledge and ideas [80]. Pharma was a partner in an industrial network for sustainable process industry firms. This network in addition to other external stakeholders were used strategically to gain legitimacy for the program and convey the relevance of EnM to the organisation by promoting Pharma's EnM at conferences, generating editorial publicity, and inviting firms, experts, academics, non-governmental organisations, and politicians to the site.

Furthermore, the energy manager's central position in the organisation allowed him to communicate directly to top management and have close relationships with operational personnel. The energy manager's position in the organisation is known to affect EnM implementation [11,40]. Personal relationships are also known to be valuable assets in innovation processes [81]. Here, the energy manager took advantage of his position in the organisation to get involved in the conceptual design of projects early on. The following quote by the energy manager illustrates how such early involvement is essential for aligning the demands, interests, and rationales of other organisational members:

> *Here, we have a project that is very good. However, it does not include that much energy saving. It is a project that is about reducing solvent in one area. It provides a yield increase and some energy savings ... I think there is more energy to be saved! Since it includes energy saving, I have worked with the concept.*

By getting involved early in the conceptual design of new projects, the energy manager uses his expertise when strategically aligning the EnM program with the organisational rationale, engaging allies, and inviting opponents to identify common goals and values. Accordingly, the analysis indicates the relevance of timing and early involvement in the translation process.

The analysis suggests that the process of translating EcoFuture into EnM practices involves selling a version of the idea, mediating others' versions, and aligning goals and agendas, thus supporting Doorewaard and Van Bijsterveld [63] who describe translation as a power-based process in which the involved actors 'continuously reshape the element of this process by confronting their own ideas with those of others and with existing organisational practices'. Furthermore, this period is characterised by the energy manager's efforts to rationalise the EnM program. Although top managers' involvement as translators was significant in the first two periods, the energy manager appeared as a key translator during this third period. The results described in this section reflects that Pharma has now reached a higher level of maturity [27,51] regarding the implementation of the EnM program and the EnM practices. Table 2 depicts the implementation process of EcoFuture in terms of translation rules and periods. The results suggest that the expected chronology in the use of the translation rules, as depicted in the conceptual framework, was not met. Instead the use of the rules had a more dynamic character. Rather than applying one translation rule in each period, the process is characterised by successive phases of translation. Within each phase, various translation rules with different intensities were applied, and different translators at various levels in the organisation were involved. We label this pattern as translation dynamics.

## 4.4. EnM Practices in Pharma

The results of the translation process are evaluated based on the premise of the materialisation of EnM practices and with reference to the theoretical 'best EnM practices' (Table 1). The identified EnM practices in Pharma are listed in Table 3. Inspired by the EnM maturity models [27,51], we have

separated into two columns the EnM practices in Pharma that are considered to comply with the best EnM practices, from those considered not-complying.

The results suggest that the EnM practices in Pharma to a large extent comply with the theoretical best EnM practice. Indeed, several EnM practices related to management, organisational routines and structures, and competences are identified in the firm. However, the analysis also reveals shortcomings in some of Pharma's EnM practices that the literature emphasises as essential. The first shortcoming concerns Pharma's use of KPIs, which for most units in Pharma do not include energy consumption or other energy related measures. Indeed, while most good management practices have beneficial spillovers on EE, an emphasis on non-energy targets is found to be correlated with energy inefficiency [24]. Hence, Pharma's design and use of KPIs might undermine the employees' motivation to improve the firm's overall EE. The second shortcoming concerns Pharma's limited integration of environmental strategies in the internal investment decision processes. Such decoupling between energy objectives and firms' investment decision processes is known as a considerable barrier for EE improvement [76,77]. Nevertheless, Pharma improved its EE considerably over the analysed period, which might seem to be a paradox. Some possible explanations will be discussed in the following section.

**Table 3.** Assessing the EnM practices in Pharma with reference to the theoretical 'best EnM practice'.

| Cat. | Best EnM Practises | EnM Practices in Pharma | |
|---|---|---|---|
| | | **Complying Practices** | **Not-Complying Practices** |
| **Management and Environmental leadership** | Top management support and awareness of energy issues | Top management awareness of and support for EnM program<br>Top managers are aware of and support environmental issues and the value of energy savings | |
| | Energy strategy (policy), planning, and targets | Long- and short-term energy reduction targets | |
| | Employee involvement, communication, motivations and incentives | Employee involvement<br>Routines to provide information about energy consumption to management and the organisation | Only production areas with a high consumption of energy have energy-related KPIs and are, hence, motivated to engage in energy-saving projects; other areas lack the same incentives |
| **Energy manager and Organisational structures** | Energy manager and the strategic positioning of the energy manager in the organisation | Allocation of resources to energy issues; two full-time employees (including the energy manager) are working with environmental reporting and energy-saving projects, with the ability to involve others when needed<br>Energy manager is positioned strategically in the organisation and reports directly to the top management | |
| **Performance measurements** | Information systems, energy audits, sub-metering, controlling and monitoring | Energy audits<br>Systematic monitoring and measuring of energy consumption<br>Systems and routines for controlling and monitoring the largest energy flows | |
| **Competence** | Staff awareness, education, and training (culture) | Internal and external EnM-related training and education programs<br>Culture of employee involvement, communication, and cooperation between departments to identify good technological solutions for reducing energy consumption | |
| **Investment decision** | Investment and pay-off criteria, and allocation of energy costs | | No earmarked investment capital for EnM; EnM projects are evaluated similar to all other projects according to compliance, HSE, maintenance, and productivity.<br>Investments in energy projects are assessed according to a short payback time of 2–3 years |

## 5. Discussion

Through the lens of the editing rules, as part of the translation theory, a picture of the implementation process of EcoFuture—from program to practice— emerges from the results (Section 4), illustrated in Figure 2 below:

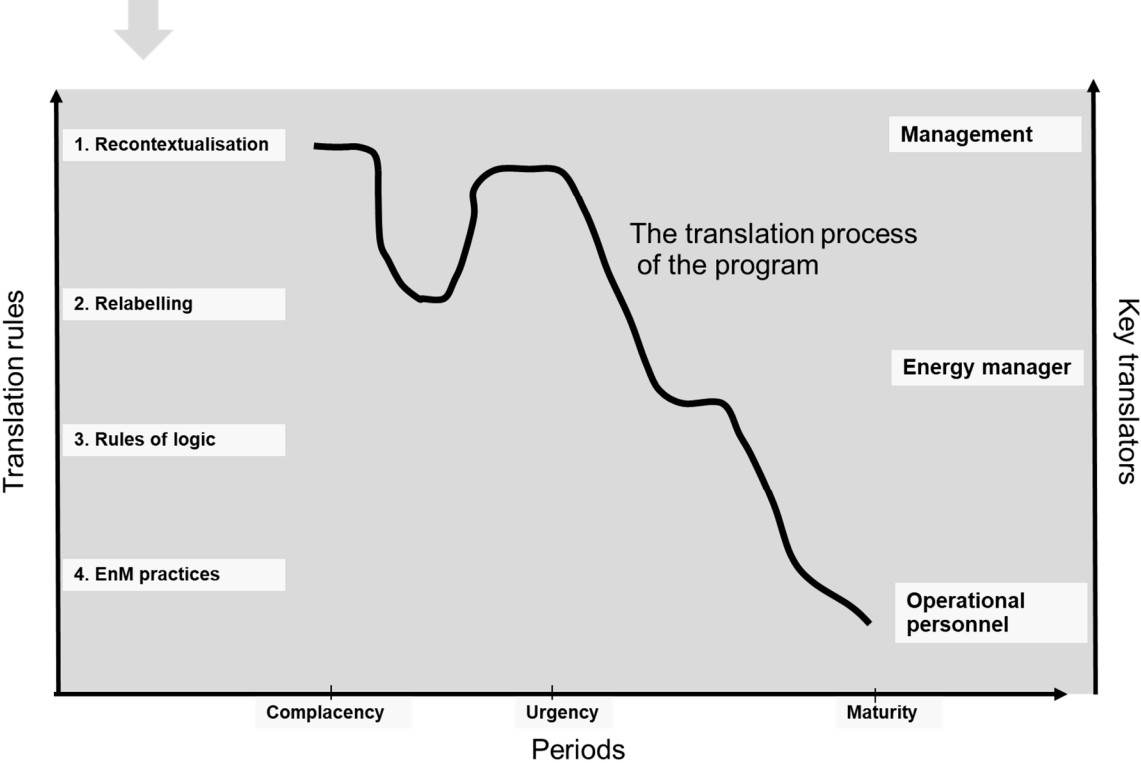

**Figure 2.** Conceptual model of the EnM implementation process from program to practise.

Figure 2 conceptualises the translation process and the resultant EnM practices in the studied firm. The figure incorporates three essential dimensions of the implementation process, namely: time, translation rules and key translators. The translation of the corporate environmental program passed through three time periods, which were recognised by changes in translation intensity, and the activation of different translators. The figure also illustrates how the translation moved back and forth over a decade, driven by different translation rules and key translators, before resulting in new EnM practices.

The results suggest that the use of the translation rules has a dynamic character. Rather than applying the translation rules successively in each period in a linear manner, the periods are recognised by different dynamics of translation. Within each period, various translation rules were applied with different intensity, and with the involvement of different translators. This finding supports prior studies stating that ideas need to undergo several cycles of translation before being applied to a new setting [56]. It also coincides with Røvik's [74] remark that ideas might alternate between passive and active phases and may linger in an organisation for a long time before materialising, leading to a gradual, slow-phased transformation of the idea to practice. Interestingly, this pattern of translation dynamics that stretches over multiple years, is an aspect that is not pinpointed in the EnM literature. Within this field, EnM implementation is usually considered from a static perspective, focusing on models and tools for assessing the end results in terms of EnM practices [38]. Moreover, the efforts and competences required to obtain to these results have received little attention [25,30]. Hence, this study

complements extant research by illustrating the complexity of implementation, and emphasising the relevance of long-time perspectives and the endurance of managers and other key translators from different levels in the organisation.

Following the analysis, we see how the top managers played an important role during the first two periods of the process. However, the energy manager emerged as a key translator when rationalising the program into EnM practices that were accepted and adopted by the operational personnel. Energy managers are often middle managers that both lack the hierarchical authority of top managers, and the immediate operational knowledge of operational personnel [59]. It is however confirmed that energy managers' operational experience [77] and position in the organisational hierarchy [11] have a positive effect on firms' EE. In line with this research, our study addresses how the energy manager used his operational experience, technical competence, social network and position in the organisation to champion the environmental program. Indeed, the results show how the energy manger worked actively, using formal and informal arenas, to rationalise energy improvements using a logic that was deemed legitimate and agreed upon in the organisation. In this manner the energy manger took the role as a change agent that 'made the change happen' [82], being translators both of the idea to be implemented and in terms of 'translating' between different levels in the organization due to their position as two-way-windows [83]. Coinciding with the increasing recognition of middle managers' role in change processes [84], our finding supports the relevance of energy managers to drive the environmental transition in manufacturing firms.

In addition to the findings that we have highlighted in Figure 2, there are two other points to be made as part of the discussion. One concerns the abstraction level of the idea, and another is the identification of additional EnM practices to those being part of the theoretical 'best EnM practices' (Table 1).

Translation scholars hold that there are variations in the abstraction level of management ideas. The abstraction level reflects whether the idea contains explicit and detailed descriptions of how it should be materialised into practices, or whether it gives room for the local organisation to make its own interpretation of the contents [62]. It is further argued that abstract ideas are more complex and thus harder to implement in a new context than those that have a more explicit and concrete character [47]. In this case, EcoFuture was oriented towards quantifiable targets without providing details on how to operationalise the program locally and is thus considered to hold a high abstraction level. The results show that the abstraction level of the environmental program allowed the translators large flexibility and freedom in the translation process, and that EcoFuture changed quite extensively when fitted to the local firm setting. In contrast to Røvik' suggestion [47], our findings suggest that the high abstraction level of the program was a success criterion rather than a barrier for the implementation of the program. As such, this finding contributes to shed some light on the challenges that firms face when they are going to implement EnM standards. A key argument in this study is that the abstract character of the environmental program implies that its contents are negotiable in each new setting. Furthermore, we argue that each local translation of the program can give rise to new versions and result in differing practices that is customised to the contextual setting of the individual firm.

The results show that most of the EnM practices recommended in the literature were put into action in Pharma. As such, the study complements prior research in emphasising the relevance of EnM practices related to environmental leadership, organisational structures and routines, and competence-enhancing activities. However, compared to the theoretical best EnM practices (Table 1), there are some vital shortages in the Pharma's EnM practices. In particular, this relates to the investment decision processes and the use of KPIs, under which environmental issues are granted limited priority. Financial limitations and firms' reluctance to prioritise EnM at a strategic level are considered substantial barriers to EE [8,40]. Despite this, Pharma reports exemplary records of EE improvements during the analysed period. Although the context bounded character of the data provided have limitations and must be closely considered, the results point to an interesting finding which we suggest is not properly captured in previous EnM studies, and which might provide an

explanation to this apparent paradox. By translating EnM according to the extant economic logic embedded in the organisation, EnM gained legitimacy and strategic relevance. Investment projects were accordingly conceptualised so that both economic and energy objectives were attained. In this way, the firm surmounted financial and organisational barriers and attained continuous EE improvements. This finding underlines the importance of translators of the environmental program working actively to align environmental and economic objectives in projects, and thereby gaining support in investment decision processes, even without earmarking funding for environmental projects. Moreover, this finding supports the claim that management ideas' capacity to travel depend on the extent to which they are associated with rational values such as renewal, efficiency, and effectiveness [85]. This way of using editing rules in translation processes invites the proposal of an additional EnM practice to the list of 'best practice EnM', one called 'translation competence' [47]. Translation competence points to the ability and knowledge of key actors and change agents to translate ideas and programs between organisational contexts. The production and diffusion of organisational ideas are important tools to transfer best practices between organizations [55,58]. Since the use of translation competence is assumed to increase the probability of achieving the desired organisational ends [43,46,47], this competence becomes increasingly important as a strategic organizational resource [47].

The current discourse in the field emphasises EnM models, and in particular maturity models [51], as efficient tools for reaching enhanced industrial EE [25,38,52,53]. These models recommend explicit EnM practices and consider them as the basic element of the analysis of EnM activity in firms [38], commonly rated as maturity levels [51,52]. Hence, the scope of these models is on the preferred end result. What is missing though in these studies, and in the broader literature, is a conceptual understanding and awareness of firms' internal processes that lead to this organisational end. In this study we demonstrate that successful implementation of EnM, in terms of the materialisation of EnM practices, is intimately connected to the organisational acceptance and preparation of the program. Furthermore, we establish the potential of the translation framework in research on EnM, by illustrating the relevance of translation rules [43,44] as a tool to guide successful implementation of EnM. Moreover, the study supports an orientation towards 'good translations' as part of these processes [44,45,59], to increase the probability of achieving more mature levels of EnM practices in the firm. Through the concept of good translation, attention is directed towards 'translation competence' in terms of managers' awareness of translation rules, abstraction level of ideas and the role of key translators. Consequently, while scholars have addressed the need for a best EnM practice [30] and EnM practice-based assessment models [38], we here propose the relevance of a 'translation framework' that complement these assessment models by supporting managers during the acutal implementaion process of environmental programs. As such, we propose a way of complementing 'result' with 'process', which can serve as reference point for both management and policy implications.

Regarding management implications, the study emphasises that organisations play an active role in translating how environmental programs are operationalised and that these processes are lengthy and often last for years. Thus, successful implementation requires managerial endurance, support, and dedication. Moreover, we suggest translation competence as part of these processes [45,46], as it may increase the probability of achieving organisational ends [47]. Managers should accordingly direct attention to good translations in terms of being aware of translation rules, the abstraction level of the idea and key translators. Furthermore, as middle managers and other employees play a prominent role during this process, managers need to encourage and educate individuals and set up organisational structures supporting the environmental change process.

Considering policy recommendation, the study suggests a positive relationship between EnM and EE in manufacturing firms and indicates that EnM is an important means to attain environmental targets. To enhance EnM practices in manufacturing firms, the results suggest that policymakers consider three mechanisms: regulation, idea complexity, and education. Policies and regulations on energy consumption and GHG emissions require organisations to adopt a concept or maintain a certain practice, as legislative compliance is a precondition for business operation [86]. Voluntary

agreements at local levels of governments are also effective in overcoming the traditional constraints of implementing top-down policies at the local level [37], and allow each firm to identify the solutions that are deemed most fitted for the local setting. The study also illustrated how the abstraction level of an idea affects the flexibility in how the environmental idea is implemented at firm level. An abstraction level that is either too high or too low might pose challenges for implementation [62]. ISO-standards [31] can for example be characterised as explicit and detailed content-wise and therefore with a low abstraction level. Hence, policymakers must be observant and ensure that the policy framework contains all relevant information required to explain and understand the EnM practices and be flexible enough to fit them into the local setting. Furthermore, this study shows that the emergence of EnM practices depends on the competence of the translators and the amount of resources devoted to educating and training organisational members. This implies that energy policies should support EnM-related education and on-the-job training.

## 6. Conclusions

This study explores the implementation process of a corporate environmental program in a manufacturing firm, a perspective that has received limited attention in the EnM literature. The conceptual framework is built on translation theory and the EnM literature, and the qualitative analysis is based on data from a pharmaceutical firm over the period 2004–2014. The analysis suggests that a wide spectrum of EnM practices have materialised during the studied period, and complements prior research suggesting a positive link between EnM practices and EE in manufacturing firms. Furthermore, this study addressed the relationship between the translation process of a corporate environmental program and the successful materialisation of EnM practices.

The results point to four main findings with theoretical relevance to the field. First, the pattern of translation dynamics illustrates how the implementation process consists of various periods of translation that evolve over time. Second, the results point to the role of the energy manager as a key translator when rationalising the idea into EnM practices. Third, the relevance of the abstraction level of management ideas is addressed. Fourth, we propose a new EnM practice to the list of 'best practice EnM', namely 'translation competence'.

Based on these findings, the study proposes avenues for future research. The translation occurs in a dynamic environment in which both the idea and context change over time [58]. Hence, it is difficult to determine whether the EnM practices emerged as a direct result of the idea or whether they would have emerged regardless of the adaptation. More research is therefore needed to obtain more knowledge on this relationship. Moreover, although this study focused on the translation process in a recipient organisation, little is known about how to effectively prepare an idea for new settings. Hence, there is need for more research about the decontextualisation phase of environmental programs—that is, translating the desired practices into an abstract representation (e.g., images, words, and texts) that is easy to recontextualise at the firm level. Such knowledge can give rise to valuable recommendations to policymakers on how to design environmental policy frameworks that can easily travel across contexts and organisations.

Furthermore, even though this study demonstrates the potential of translation theory in research on EnM, we have only applied some selected elements from this rather mature framework which has gained attention and attracted different fields [41]. Due to the pressing relevance of sustainable management programs and sustainable business models, the development and diffusion of organizational ideas has become increasingly important for businesses as a strategic organizational resource, for research and for the larger society. Hence, we suggest that EnM scholars should explore further the potential within this theoretical framework to further conceptualise translation competence as an EnM practice and develop best translation models of EnM.

**Author Contributions:** Writing—original draft: M.T.S.; writing: review & editing: M.T.S. and E.A.N. Both authors have read and agreed to the published version of the manuscript.

**Funding:** This research received no external funding.

**Acknowledgments:** The publication costs for the article are funded by the publication fund of UiT The Arctic University of Norway.

**Conflicts of Interest:** The authors declare no conflict of interest.

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
