# Peer review of "From Program to Practice: Translating Energy Management in a Manufacturing Firm"

_sustainability, doi:10.3390/su122310084_

Round 1

Reviewer 1 Report

-The overall level of the paper is good: even if it is quite simple, it is well written and some important considerations are highlighted. The Introduction and Background sections provide useful information for the readers. Nevertheless, some information presented is not accurate. The author also needs to justify the importance of understanding the vision of the topic. The motivation of study thus needs to be re-written to show stronger linkage. It would be nice to see a stronger connection between your findings and the theme of the journal. How does this understanding help organizations and the industry to make better decisions? The managerial and theoretical implication is missing at the moment. Thus the reviewer cannot conclude if the manuscript has contributed fully to the existing body of knowledge.

- The style of introduction must be coherent, and it should explain what the problem is, what has been researched in previous academic literature in this area, and what actually gap exists .further, how this study fulfills this research gap. I  suggest revising the whole introduction part, and provide what previously has been researched in this topic and what is the research gap?

-I suggest rewriting the last paragraph of the introduction. The three contributions are not well articulated in the paragraph.

How do technological innovation and fiscal decentralization affect the environment? A story of the fourth industrial revolution and sustainable growth

https://doi.org/10.1016/j.techfore.2020.120398

Role of Design Thinking and Biomimicry in Leveraging Sustainable Innovation

https://doi.org/10.1007/978-3-319-71059-4_86-1

Steering for Sustainable Development Goals: A Typology of Sustainable Innovation

DOI:

 https://doi.org/10.1007/978-3-319-71059-4_64-1

- The overall quality of the paper is good, but according to my opinion, you must restrain this article to only one research question. It is difficult to understand the aim of the study, as also the structure is almost confusing. My opinion is to combine all questions into one question where the author mentions y over a period of 10 years ( see page 2, Line 92).

-Moreover, the literature cited is not exhaustive and is relatively not recent. You might want to increase the number of works cited, explain in detail what they find and what are their actual limitations. Moreover, you should add more recent studies on the topic. A more accurate way to structure the theoretical section would be to separate the theoretical framework and the literature review from the hypothesis development.

Exploring the effect of buyer engagement on green product innovation: Empirical evidence from manufacturers,

DOI: https://doi.org/10.1002/bse.2631

Benefits of cross-border Collaboration at the Base of the Pyramid markets for innovation improvement

https://doi.org/10.4324/9780429424151

Industrial ecology in support of sustainable development goals.

 DOI: https://doi.org/10.1007/978-3-319-71062-4. 978-3-319-71062-4

Effects of buyer-supplier relationship on social performance improvement and innovation performance improvement

https://doi.org/10.1504/IJAMS.2019.096657

-It is very unclear to me the methodology that authors take into consideration. This is an important point before I can start to reason about the result analysis. Does the reader need to understand precisely if the proposed methodology can answer the research problem?

- I think the authors are not sure about the methodology, and it is a serious concern for the result's reliability. Because the chosen method is dubious, and it could derail the reader. Indeed, authors have performed a good methodology.but in my opinion, in quantitative studies, the reliability of the data is most important for the conclusion.Why authors selected this data collection methodology?

-The provided data analysis results are not completely convincing to me, again, they are too vague and generic. Again, I suggest taking inspiration to the following paper to explain, where it could be applicable.

Minor General Comments

- The manuscript is potentially original contributive but needs a minor revision.

-Some sentences from the conclusion could be moved up in the discussion section. A conclusion section must be well written and clearly explain the study findings.

- Implications for future research may also be included in the conclusion at the end. This research has article has created a lively discussion on so many issues that were hitherto unheard of and not addressed.

- How the results of this study can be generalized to other companies.

-- Also explain briefly what the future research opportunities are.

Author Response

Thank you for your helpful and constructive comments to our manuscript, we do our best to address your concerns in the attached document

Reviewer 2 Report

The topic is actual and fit to the themes of journal. There are some minor corrections Author should make before final submission. The paper can benefit from these minor revisions. Comments and recommendations related to the article are the followings:

  • The paper is well-written, understandable and clear, it fits the aims of the journal. The study explores an interesting and under-researched area of knowledge. The article closely corresponds to the topic specified in the title. 
  • The literature review is well structured and refers to an appropriate breadth of resources. However, it is suggested formulating clear research questions and describing research objectives more clearly in the Introduction section. 
  • The methodology could include a little more detailed information regarding how the data were collected, how the questionnaire was administered, how many respondents were asked and if there was any selection process of the sample.
  • There are a few minor grammatical/formal errors which need to be corrected but the cohesion of communication is clear and concise - for example: In text citation should be revised (in case of more than 2 authors "Last name of first author et al." should be used instead of listing all authors eg. Shulze et al.)
  • Since the paper is mainly a describing study, it is recommended to use more critical commitment or opinion expression. Author should highlight the differences among the diverse approaches more definitely - which approach can be accepted and agreed by the author, which are not and why.
  • In results and discussion sections (when examining each period) the Author adequately reflects on the framework model set up on the basis of the secondary research primarily the literature review. It can be clearly seen the most important correlations between the theoretical framework and research results. However, it is suggested highlighting the most relevant findings in a separate Conclusion section. It is recommended to emphasize the benefits and contribution of this current study to the knowledge of EnM adoption and implementation process (especially within MNCs). Conclusion section should include the possible research directions in the future which are more or less detailed in different parts of the study.

Author Response

(The authors gave the same response as above.)

Round 2

Reviewer 1 Report

- The authors have addressed the majority of my comments at the previous round however, there are still some slight corrections that I recommend making before accepting the paper for journal publication: - The topic of the paper is potentially interesting. However, I think some several critical points and weaknesses impede to publish the paper.

-In line 39, where author discuss technological innovation, subsequently, I sugget author do describe the technological innovation role. Here I recommend a recent publication.

How do technological innovation and fiscal decentralization affect the environment? A story of the fourth industrial revolution and sustainable growth

https://doi.org/10.1016/j.techfore.2020.120398

-Following, line 4, page 2, when author discuss the energy management, I suggest aughors to continue the paragraph by highlighting the importance of energy management.For example,in a study, author argue the importance of energy management for the sustainable development.

"Sustainable Development through Energy Management: Issues and Priorities in Energy Savings." Res. J. Appl. Sci. Eng. Technol 7 (2014): 424-429.. http://dx.doi.org/10.19026/rjaset.7.271

-Especialy in the introduction, I suggest authors to link what sustainability is and how it support the study objective. For the improvement of the paper, I may suggest to incorporate.

-I appreciate the authors for  good work. Heding 2.1, require to little explain about product and process innovation. How environmental management support it. For example.

Steering for Sustainable Development Goals: A Typology of Sustainable Innovation

DOI:

 https://doi.org/10.1007/978-3-319-71059-4_64-1

Progress from Blue to the Green World : Multilevel Governance for Pollution Prevention Planning and Sustainability.

https://doi.org/10.1007/978-3-319-58538-3_177-1

-Some sentences from the conclusion could be moved up in the discussion section. A conclusion section must be well written and clearly explain the study findings.

Author Response

Please find attached our response to your comments 
